# How the Covid-19 Pandemic Influenced the Approach to Risk Management in Cycling Events

**Filippo Bazzanella** [1,*], **Nunzio Muratore** [2], **Philipp Alexander Schlemmer** [3] **and Elisabeth Happ** [3]

1   Department of Strategic Management, Marketing and Tourism, University of Innsbruck, Karl-Schönherr-Straße 3, A-6020 Innsbruck, Austria
2   Luiss Business School—Management School, Villa Blanc, via Nomentana, 216-00162 Rome, Italy; nmura90@gmail.com
3   Department of Sport Science, University of Innsbruck, Fürstenweg 185, A-6020 Innsbruck, Austria; Philipp.Schlemmer@uibk.ac.at (P.A.S.); Elisabeth.Happ@uibk.ac.at (E.H.)
*   Correspondence: Filippo.Bazzanella@student.uibk.ac.at

**Abstract:** The COVID-19 pandemic has taught us to live in social isolation and has brought an important element of social life, the events industry, to a complete standstill. In resurrecting the events industry, the most urgent focus is on managing the risk of any crowd-control measures with a view to reducing to zero the danger of the virus spreading. This research focuses on the main issue of the impact of the coronavirus disease 2019 (COVID-19) on the organization of sports events (SEs), and in particular, cycling competitions. This study, therefore, aims to provide deeper insights into (a) the measures introduced to face the health emergency situation in cycling events, (b) the comparison of these measures with previous experiences in similar SE contexts, and (c) the possible evolution of organizational models for cycling events in the post-pandemic era. Fifteen semi-structured interviews with cycling athletes, managers, and officials constitute the methodological basis for this study. The results show that countermeasures have been taken that are effective in dealing with pandemic characteristics and are likely to be applied in the future, while others will be phased out or used again only when necessary. This study enhances scientific knowledge by analyzing a renewed approach to risk management for SEs, with a specific focus on pandemics and medical risks. Finally, the study shows that cycling events need to adapt the specifics of such a new approach to the standards projected on future scenarios for which the COVID-19 pandemic has paved the way.

**Keywords:** sports events; Covid- 19; risk management; SARS-CoV-2; cycling events; organization; crisis

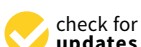



## 1. Introduction

The biennium 2020–2021 will be remembered as a period that marked history, and as a watershed moment for all humanity. The coronavirus pandemic 2019 (COVID-19), the first in recent history, has subjected the world to a transversal shock affecting all aspects of human actions and forcing humanity to fundamentally rethink or eliminate previous behavioral patterns and models (Westmattelmann et al. 2020). The social distancing imposed by the rapid and dangerous spread of COVID-19 has not spared sport. The new coronavirus has paralyzed the entire world of sport, causing a complete global halt to sports competitions at all levels, including the XXXII summer Olympic Games in Tokyo, which represent the maximum expression of sport and its values (Patsantaras 2008). This case, as well as other major mass events, including cycling events, have been hit hard by this sudden wave.

Guaranteeing the safety and surveillance of major SEs has become considerably more complex (Giulianotti and Klauser 2010). In the past, various "dangerous" events (Blumberg et al. 2016) have indeed impacted the organization of SEs. They have however, never reached the gravity and scope such as that triggered by COVID-19.

Major cycling events (professional, amateur and touristic) generally adopt the approach of combining the participation of large groups of people with the involvement of the media and other stakeholders (Bowles et al. 2006; Pelzer 2010). It is, therefore, necessary to consider the impact of events on tourist destinations (Chalip and McGuirty 2004) and in particular the effects of possible cancellations or changes in management, for example, due to the sudden onset of a pandemic. After the first global lockdown, some events, and in particular the American professional leagues (Dove et al. 2020) immediately organized themselves to ensure that they would not miss the entire 2020 season; "buffer" solutions were put in place which cannot, however, be seen as a definitive solution even in the post-pandemic period (McHill and Chinoy 2020). The scientific literature on risk management and mitigation in relation to SEs (Fuller and Drawer 2004; Leopkey and Parent 2009) is facing a new challenge with the global pandemic as a new type of global health risk has appeared in the SE system.

Taking a closer look at cycling events in terms of the pandemic, scientific literature has proposed so-called protective "bubbles". These measures effectively mean that the subjects involved in the event (teams, staff, etc.,) are separated in groups that are continuously monitored and isolated from external contacts (whose health is neither controlled nor necessarily safe) during the course of the event, forming a large "bubble". Although the bubble system has produced good results, it has not managed to completely eliminate the risk of contagion with COVID-19. During the Giro d'Italia 2020, various teams were forced to withdraw after numerous participants tested positive for Covid-19 (Hytner and McLaughlin 2020). This has seen the creation of a very important and new event role known as "COVID-19 Coordinator". This operative takes on fundamental responsibilities in the first organizational phase, is an expert in infectious diseases and is designated by the Local Organizing Committee (LOC).

Although more research has been carried out into the security governance of SEs in recent years (Ludvigsen and Parnell 2021; Whelan and Molnar 2018), gaps remain in risk-management approaches especially in relation to health risks (Ludvigsen and Hayton 2020). We have, therefore, conducted an investigation into the organizational risk management of SEs with a specific focus on the effects of the pandemic emergency on the organizational processes involved in cycling events (professional and amateur). By means of qualitative semi-structured interviews, we aimed to address the following research questions: (a) What measures have been introduced to tackle the health emergency situation at cycling events? (b) How do the new measures differ from previous experiences in similar SE contexts? and (c) What is the possible evolution of organizational models for cycling events in the post-pandemic era?

Several factors, interests, and stakeholders are affected by the cancellation or postponement of an event and it is the task of the experts to balancing them to find a way out of this blind ally. Studying the application of risk management in the context of SE organization (Fuller and Drawer 2004; Leopkey and Parent 2009), therefore, provides a basis for mitigating potential risks when staging outdoor, dynamic cycling events in the new post-pandemic age.

## 2. Theoretical Background

### 2.1. Literature Review

Event studies have evolved as an important area of research in recent years, drawing on several foundational disciplines and constituting an interdisciplinary field of research (Bowdin et al. 2011; Getz and Page 2020). A more specific scientific niche considers the phenomenon of event tourism, which attempts to interpret definitions, managerial aspects, and implications (Getz 2013). Some authors (Getz 1998; Hall 2001; Higham and Hinch 2003) have also defined the meaning of SEs with a view to giving a clearer and more complete description, distinguishing between recurring and one-off events etc. Gratton et al. (2000) as well as Barget and Gouguet (2007) identify different types of sports events, differentiating them on the basis of certain characteristics.



Until the outbreak of the COVID-19 pandemic, SEs played a major role on a global scale. Assessing from different angles, the impact and multiple implications of SEs have been the subject of several studies (Parent and Smith-Swan 2013; Preuss 2013; Masterman 2014).

The issue of risk management at SEs has become increasingly important, both from a theoretical point of view and in terms of the managerial implications (Leopkey and Parent 2009), especially following the terrorist attack of 11 September 2001.

For the purposes of this research, it is crucial to have a complete overview of the ways in which risk management is evaluated by the organizers of SEs as well as the many other stakeholders.

In addressing the issue of risk in major sporting events, we have often found a link to the legacy of the event (Parent and Smith-Swan 2013; Whelan and Molnar 2018). Especially in the context of major events, the issue of legacy plays a substantial role (Preuss 2015). The literature review by Thomson et al. (2013) provides clarity regarding the different definitions. A clear development in recent decades has been the growing concept of environmental, social, and economic legacies (Chappelet 2012; Minnaert 2012; Preuss 2015). To understand the importance of risk management in relation to the legacy of SEs, it is essential to comprehend what the term actually means. In his definition of legacy, Chappelet (2012) adds the dimension of intentionality.

If we consider a negative legacy, we can assume it is the result of intentionality but looking at the six dimensions that Preuss (2015) illustrates, we can assume that a negative SE legacy has consequences for multiple stakeholders. Consequently, it is necessary to extend the concept of legacy to leverage (Chalip 2004) and define it as all actions aimed at optimizing the results of a positive legacy or at least reducing the risks of a negative legacy (Parent and Smith-Swan 2013).

Risk management at events is a relatively well-researched topic. An essential contribution to the understanding of this sensitive subject comes from the studies included in the Event Management Body of Knowledge (EMBOK), where risk analysis is found within the processes identified in the EMBOK model (Rutherford Silvers 2008; Bowdin et al. 2011; O'Toole 2011; Goldblatt 2014). Berlonghi (1990) assumes that the objective of risk management is to prevent and even eliminate loss or damage by "making events as safe and secure as possible".

Among the many risk factors taken into account by recent studies (Rutherford Silvers 2008; Leopkey and Parent 2009), this research focuses on the medical emergency caused by the COVID-19 pandemic. In Table 1, we note that the literature to date has underestimated the pandemic's impact on event-related risks. A pre-pandemic study pointed out that "risk management is central to protecting health and safety" (Windholz 2016), encouraging the careful observation of regulations at all levels by event organizers, working in close cooperation with all event stakeholders to develop a model that can facilitate the effective sharing of risk-management responsibilities according to each stakeholder's ability.

The COVID-19 pandemic, which is still ongoing on a global scale, has nevertheless allowed us to take a look at some recent studies (Barbosa et al. 2020; Garcia-Garcia et al. 2020; Schnitzer et al. 2020; Wong et al. 2020; Rico-González et al. 2021) that offer some analysis and provide partial answers or solutions. Initial recommendations by Wong et al. (2020) included measures to control local spread through public awareness, the encouragement of personal hygiene, and the postponement or cancellation of large-scale public events. This was the case, for example, for the Tokyo Summer Olympic Games, originally planned for the summer of 2020. There is also an "ethical" risk to be considered, another consequence of COVID-19. This is the fact that many athletes have completed or are finishing their suspensions for doping and will be able to participate in all those competitions postponed because of the pandemic (Garcia-Garcia et al. 2020).

**Table 1.** Elaborated summary of previous event-related risk and risk-management issues highlighted in the scholarly literature (Leopkey and Parent 2009).

| Author | Risk Issues/Areas/Topics |
|---|---|
| Chang and Singh (1990) | People (e.g., employees, athletes, volunteers)<br>Public (e.g., spectators, local community)<br>Property (e.g., equipment, facilities)<br>Financial/legal risk<br>Safety and security (e.g., physical hazards)<br>Television revenue risk (i.e., loss of revenue)<br>Political (e.g., international terrorism, threats, demonstrations) |
| Getz (2005) | Financial risk (e.g., loss of revenue, theft, legal issues, unanticipated costs)<br>Management risk (e.g., goal displacement, takeovers, management failures)<br>Health and safety hazards (e.g., accidents, medical issues, threats, emergencies)<br>Environmental risk (e.g., pollution, natural disasters) |
| Frosdick and Walley (1997) | Spectator/Crowd risks (e.g., poor seating, ticketing issues)<br>Commercially related risk (e.g., to advertisers and sponsors)<br>External disruption risk (e.g., noise and projectiles from event)<br>Safety and security risk (e.g., venue condition) |
| Bjarnason and Cannell (1999) | Workers, proper documentation, good communication practices, valid insurance, secure facilities, equipment, emergency medical services, action plans |
| Chappelet (2001) | Corporal risk (e.g., related to quality and density of people)<br>Material risk<br>Environmental risk<br>Fraud risk<br>Meteorological risk<br>Image/public relations risk |
| Peterson and Hronek (2003) | Nature<br>Human incidents (e.g., crime, vandalism, hooliganism, terrorism) |
| Appenzeller (2005) | Ticket sales, sponsor services, athlete services, hospitality, operations, concessions, support services, advertising, promotions, media relations |

Today, we are in the middle of the pandemic, and major events are still applying contingency plans by trying out new "temporary" organizational models. Public opinion sees vaccines as a possible solution to this pandemic. However, set in a different historical context, Parent and Smith-Swan (2013) addressed the question of whether imposing the vaccine on all participants and stakeholders of a sporting event would constitute a violation of their personal freedoms. As Ludvigsen and Hayton (2020) claim, it is very difficult at this stage to give concrete answers to the many questions relating to risk management which have been raised by the COVID-19 pandemic among scholars and practitioners involved in sporting events. The present study is likewise intended to offer a summary and representation of the stakeholders' point of view in light of recent discussions that have implications for both researchers and professionals.

### 2.2. Past Cases of Dangerous Diseases Impacting on SEs

Recent experiences of mass SEs that have taken place regularly, successfully, and safely, despite the presence of serious and global health emergencies, can be found in the literature (McCloskey et al. 2020): the Winter Universiade in Serbia during the 2009 Influenza A (H1N1) epidemic (Loncarevic et al. 2009), the SEs held in Africa during the Ebola epidemic between 2014 and 2015 ("African Youth Games" in Botswana, "All Africa

Games" Republic of Congo, "Africa Cup of Nations" in Equatorial Guinea) (Blumberg et al. 2016), and the 2016 Rio Olympics threatened by the spread of the Zika virus.

The approach adopted to manage the health risks at these events can be taken as a useful reference for the purposes of comparison with the COVID-19 pandemic. It is interesting to note how the events took place in safe conditions in all these cases thanks to a combination of several factors (Loncarevic et al. 2009; Blumberg et al. 2016):

- Adequate prevention and information campaigns in the months before the event.
- Effective coordination of resources and authorities: creation of specific crisis management committees and teams of doctors and expert operators working 24/7 throughout the national territory.
- An effective system of alerting and tracking travelers and athletes, operating before and during the event.
- Precise and clear rules for the treatment of suspected and positive cases with the purpose of isolating infected people in specific and equipped areas at the event sites.

These measures were successful in containing the spread, thus avoiding the cancellation of the aforementioned events. In Serbia, 13 cases were recorded before the event, due to returning travelers. At the end of the Games, 7 cases were confirmed (six athletes and a volunteer). The clinical history of these cases showed that only four of them were attributable to a contagion that occurred during the event. Even during the Rio Olympics, the number of infections relating to foreign delegates was close to zero. Several studies (Rodriguez-Valero et al. 2018) that analyzed the phenomenon, claim that the causes of minimal spread can be found not only in environmental factors (as the season was not favorable to mosquitoes), but also in the measures adopted to control and track the suspected cases and possible symptoms. In particular, it was highlighted that adequate countermeasures against insect bites, correct eating habits, correct management of the delegates' movements, and the reports held by the delegations allowed infections to be contained, thus guaranteeing adequate levels of safety during the sports competitions. Accurate information and awareness, combined with the timeliness of the intervention, therefore proved to be decisive in ensuring that the event and the economic, sports, and social interests associated with it; were not compromised (Vancini et al. 2016).

These cases show how an emergency can be managed with an adequate mitigation plan that includes targeted prevention, accurate identification of the risks and custom-fit methods and measures. An effective risk-assessment and management model in each of these cases, therefore, meant that the SEs were able to take place in acceptable safety conditions.

### 2.3. Project Management and Risk Management in SEs

SEs are organizational challenges characterized by complexity, time restrictions, and their predetermined life cycle from start to finish (Parent and Ruetsch 2021).

As a complex and multi-relational context, the interests of many stakeholders (for example, athletes, officials, volunteers, spectators, authorities, sponsors, residents, and media) need to be considered conjunctly (Cuskelly et al. 2006). The creation of sports events is a natural field in which project management methods and techniques are essential in the definition and planning of the different phases (Cserháti and Szabó 2014; Pielichaty 2017). Goldblatt (2014) highlighted the importance of applying project management theories and techniques to the event sector mainly to define the processes and objectives of the project itself, whereby the utilization of a project management system helps to establish a systematic approach to all kinds of events. Successfully managing the complexity of a sporting event entails identifying and dealing with potential risks that may threaten the event. For the development and success of a sporting event, the application of respective risk-management techniques is decisive (Spengler et al. 2006).

A typical risk-management process therefore encapsulates two main phases, one strictly prodromal to the other: (1) the risk-assessment phase (Fuller and Drawer 2004; Rutherford Silvers 2008); (2) the subsequent risk-treatment phase—actual risk management

(Rutherford Silvers 2008). The first phase is the heart of the entire process, which aims at focusing on the approaching risk situation and at defining the organization's position toward internal and external risk factors. The risk-assessment phase is characterized by its high level of technicality. Once this phase has been completed, the risk treatment is carried out. The second phase is characterized by decision-making—based on the results of the risk assessment, specific actions and measures are selected and implemented with a view to managing the risk (Rutherford Silvers 2008; Leopkey and Parent 2009). The impact assessment of the implemented measures will produce a residual risk-assessment phase where a new risk estimate is made. The process is completed with a residual risk-reporting phase.

The process of risk management can be implemented as part of a best practice management system within the sports and leisure sector (Fuller and Drawer 2004). In the organization of SEs, the outlined process is applied to various cases (Parent and Smith-Swan 2013). In particular, the macro areas of risk identified in the scholarly literature can be traced back to the following: (a) legal risks, (b) risks to health and safety, (c) compliance risks, (d) decision-making risks, (e) security risks, insurance risks, and (f) risks and emergency management. For the purposes of this work, it is the health risks that must be investigated.

In terms of cycling events, the COVID-19 pandemic has forced the complete rescheduling of national and international fixtures generating effects that have never been seen before (Grix et al. 2020). The Tour De France has maintained and strengthened its role as a benchmark for the entire system, having been delayed only by a few weeks. All other events on the "UCI World Tour" circuit have, however, been rescheduled and moved to fall 2021. These measures have generated a unique temporal overlap of events that are traditionally located at very different times of the year, and which strongly shape their identity. This new perspective represents a challenging testing ground for the world of sports management. Numerous factors, interests, and stakeholders are affected by the cancellation or postponement of an event.

Based on the current circumstances, this study gives detailed insights into the application of risk management in SE organization (Fuller and Drawer 2004; Leopkey and Parent 2009) which is a requirement for mitigating the risks of outdoor and/or dynamic cycling events at their source in the new pandemic scenario. The objectives of the study at hand can, therefore, be summarized by (a) focusing on the measures introduced to face the health emergency situation in cycling events, (b) comparing these measures with previous experiences in similar SE contexts, and (c) the possible evolution of organizational models for cycling events in the post-pandemic era.

## 3. Methodology

### 3.1. Overview

In order to assess the impact of a complex phenomenon, such as a pandemic, it is necessary to analyze the concrete effects it has produced and the countermeasures put in place, so that possible applications and future consequences may be deduced. Starting with the current Covid-19 pandemic, this work aims to investigate and explore possible future scenarios. We adopted a combination of an inductive and a deductive approach, where we related codes (categories and concepts) to each other (Mayring 2014). The lack of historical data relating to phenomena of the same type and scope led us to adopt a qualitative approach by conducting semi-structured interviews (Ciucci 2012). Ritchie et al. (2014) stated that one of the most distinctive features of qualitative research is that the approach allows issues to be identified from the perspective of the study participants while gaining an understanding of the meaning and interpretations that they give to behavior, events or objects. Furthermore, the use of a qualitative social research technique, based on a broad-spectrum vision, allows the problem to be grasped not only in its objective dimension, but also in terms of the impact it has on relationships between subjects and stakeholders (Ritchie et al. 2014).

### 3.2. Research Setting, Participants, and Procedure

In an effort to find answers to the three research questions—(1) What measures have been introduced to address the health emergency situation at cycling events? (2) How do the new measures differ from previous experiences in similar SE contexts? and (3) What is the possible evolution of organizational models for cycling events in the post-pandemic era?—fifteen experts in the field of cycling events (the selection process was reviewed by two experts) participated in guided semi-structured interviews in person (data collection from June 2020 to October 2020). We adopted a procedure for recruiting respondents (Rapley 2004) that followed the snowball technique (Scott 2000). In Appendices A and B, we have detailed information concerning the questions and samples. The research area was located in Italy: all the experts interviewed were Italian and the cycling events studied took place in Italy during the 2020 season. To create an adequate research sample, the profile of the interviewees was selected to include a full spectrum of experiences, several fields, and different professional figures: from parts of the National Federation to event managers and ex-professional riders. The most frequently represented professional area was "Organization Management" with seven people interviewed.

Overall, the interviews comprised ten questions and covered the following topics: (1) event management (four questions), (2) risk management (three questions), and (3) sport tourism (three questions). All questions were designed as open questions allowing a wide range of personal answers. Appendix A provides an overview of the interview guideline.

The average length of the interview was over 40 min. Different contact tools were used (Skype©, phone, Whatsapp©, written form).

In terms of process validation, two of the authors had experience in sports event management and, therefore, evaluated both the interview layout and the content validity. The sampling technique started with a list of people who were considered to be significant. During the course of the interviews, respondents were asked to name other people they considered important for investigating the research topic. The authors adopted the concept of saturation (Glaser and Strauss 1967; Guest et al. 2006), and the sample was completed when the last person interviewed named other people who had already been interviewed (Appendix B).

After interviewing the participants, the interviews were transcribed in Italian and then translated into English. Each session was digitally recorded and transcribed. To ensure the quality and reliability of transcriptions and translations, a professional language editor fluent in English and Italian was consulted during the translation process. Subsequently, the data were analyzed on the basis of qualitative content analysis (Mayring 2014; Neuman 2003). The data analysis led to categorization through an open-coding process. A member of the research team coded the transcripts using Microsoft Excel© (v. 16.50). Afterwards, the authors compared codes, discussed the patterns in the data and identified the main themes that emerged from the responses given in the interviews. The following five categories (macro areas) were extracted: (1) type of event, (2) level of activity, (3) stakeholders, (4) measure of mitigation, and (5) future applications. The aim of the macro-area filter was to identify the principal and relevant information linked to the research questions. Within these five macro areas, additional macro-categories and sub-categories were identified (moving from general to particular) for a more accurate classification. An identification code was then assigned to each macro-area in order to highlight, in the transcription of the interviews, the relevance of the data obtained from the interviewee in relation to a specific thematic area. Table 2 provides an overview of the macro-areas, macro-categories, and sub-categories. The relevant sections of the interviews, analyzed with the three-layer filter scheme, were put into these categories to arrive at a final result by paraphrasing and summarizing the interviewees' answers. The most frequent and relevant concepts and opinions were reported in interview quotes to highlight the most critical points and findings of the analysis.

**Table 2.** Processing data filters.

| Macroarea | Type of Event (Code MA1) | Level of Activity Practice (Code MA2) | Stakeholders (Code MA3) | Measuring Mitigation (Code MA4) | Future Applications (Code MA5) |
|---|---|---|---|---|---|
| MACRO CATEGORY | Single day, stage races | Professional, amateur, tourist | Sponsors, institutions, voluntary associations | Impact on organizational costs, degree of effectiveness in containing infections | Measure capable of lasting in the future, measure abandoned after the pandemic |
| SUB CATEGORY | Three-week major tours, minor stage races, one-day races | High-level professional competitions, youth competitions, territorial promotional events | visibility in the media, Opportunities to meet and network at events, contact with fans | Measuring mitigation on the competition site, for the athletes, the organizational staff, and the host structures | Standard measure for all events, suitable and effective only for some |

Through the multiple points of view that emerged from the professional areas analyzed and the different levels of activity explored (professional, amateur and touristic), it is possible to gain a panoramic view of the first Covid-19 response in the world of cycling events.

**4. Findings**

The interview outputs show the main critical issues encountered in dealing with the pandemic emergency. The varied spectrum of subjects and professional areas represented provide a general overview of the effects of the pandemic on the "cycling system". The opinions of the interviewed stakeholders produced general agreement on the main actions and measures adopted. Among the common elements found, there was clearly a lack of certainty and secure prospects in the short and medium term.

The following summarizes the main outputs of the interviews for each macro area:

Macro area 1: Type of event

The first macro area aims to understand the possible and specific problems COVID-19 has generated, classifying them on the basis of event type: one-day races and stage races.

In the case of one-day races, it is easier to manage the event and the people around it. By comparison the time dilation typical of stage races represents a risk factor and considerable organizational complexity that could have a great impact for the organizers. Creating and maintaining isolation for hundreds of people for three weeks significantly increases both cost and responsibility. These conclusions were confirmed by the testimonies of the interviewees:

> "Stage races require greater measures and controls, compared to the past—the athletes were isolated with single rooms and all the equipment was continuously sanitized". (I-13)

In terms of the organizers' responsibilities, it also emerged that the prevention of infections cannot extend beyond the typical start and finish areas of the race or stage:

> "The division into areas of the site of race, however, excessive responsibilities cannot be placed on the organizer outside—these spaces: start area, finish area". (I-2)

A further difference to be found with respect to the type of race relates to the award ceremonies. In the case of stage races, they tend to be longer due to the different jersey rankings that often change many times during the event.

In general, these procedures have been simplified and carried out in accordance with the anti-contagion rules provided by the protocols.

Macro area 2: Level of practiced activity

In the second macro area, the information was classified by examining the effects of COVID-19 on the practice of the sport, based on the level of activity carried out. This investigation perspective was necessary because the needs of professional cycling events are very different from those of amateur cycling and cycle-tourism.

The opinions of the interviewees reveal a shared and general concern for the increase in costs and a reduction in economic resources. These concerns are even stronger in relation to youth activity:

> "I unfortunately believe that youth cycling, which does not have the same resources as professionalism, will be heavily penalized by the reduction in funds and the number of racing". (I-10)

The high-level professional system, though in difficulty, represents an elite of the movement which, as such, enjoys the greatest attention and protection. The youth movement, on the other hand, does not have comparable resources and is often entrusted to organizers moved more by passion than by concrete economic returns. There is a real risk that an entire generation of athletes, potential professionals, and future champions, will be compromised.

Among the insiders, there is also a common desire not to distance cycling, a popular sport par excellence, and separate athletes from its fans. While respecting the rules and protocols, this factor is considered to be of great importance:

> "Cycling is a popular sport that brings champions close to the people, this element must not be lost even in this harsh reality that COVID-19 requires us and, above all, in the future". (I-15)

The amateur sector, has seen the spread of alternative event formulae, which are simpler and come with fewer competitive connotations.

> "I am in favor of alternative event formulas, at an amateur level, but without distorting the typical and most exciting elements of cycling as a sport". (I-7)

The most enterprising and courageous organizers have chosen, where possible, to stage their events by implementing formats that are compatible with social distancing rules even if it means welcoming a smaller number of members and being burdened with higher costs.

In terms of touristic cycling activities, there is general consensus that cycling is an excellent means of promoting a territory, while encouraging economic growth and social cohesion:

> "I believe that the sport tourism impact is fundamental aspect of a sporting event, we must not underestimate but rather seek strongly, the link and synergies with local administrations". (I-14)

Many of the interviewees stressed that, in the absence of competitions, new life can be breathed into this sector—allowing the safe practice of sports while safeguarding the economic benefits for the sporting event's host location. In this sense, great importance is given to the ability of the accommodation sector to deliver:

> "I believe that the development of sports tourism is closely linked to the adequacy of accommodation facilities and tourist staff ( . . . )" (I-9)

Macro area 3: Stakeholders

Any organization relates to external subjects capable of influencing their own assessments and choices. Stakeholders represent a fundamental variable for the organizer of a sporting event. By virtue of this assumption, the third macro area refers specifically to all the implications that the pandemic has had in terms of the relations between the various subjects who have economic or other interests in the event.

A leading role is played by sponsors and commercial partners: professional teams as well as race organizers draw most of their resources from sponsorships. The interviews revealed a state of general suffering, as the economic crisis caused by the lockdown has forced companies to revise their spending budgets:

> "We experienced dramatic moments during the lockdown, cycling teams get the most of their income from sponsorships and in a time of crisis these are the first costs that companies cut, especially if they are not able to generate significant visibility". (I-12)

However, the sponsors who have made important investments in events (especially professional ones) have not failed to fulfill their obligations, demonstrating how cycling, even in a moment of crisis such as the present one, remains an attractive stage for companies. In this sense, the emphasis was placed on visibility:

"The absence, or limited spaces, of the exhibition and meeting areas within the events, due to the rules of distancing, I believe it will come to a cut in sponsorships—given the lower visibility offered, which must always be guaranteed as much as possible, and to the possibility of networking at events which cannot be completely eliminated". (I-8)

Without this dual aspect, there is a risk of investors fleeing:

"I believe that even in the presence of limitations, teams and companies must be guaranteed opportunities for networking during events: the development of business to business relationships is fundamental for the economic resources of the sponsorship system". (I-13)

A positive factor is also to be found in the renewed confidence among the sponsors in the sector, which has seen no significant decreases:

"Fortunately, however, many companies remained because being linked to the sector, they had a direct interest strategic to maintaining the partnership". (I-12)

Various subjects interviewed affirmed that, in terms of commercial partnerships and sponsorships, cycling would need to develop new forms of entertainment with innovative products, television and multimedia content to attract investors.

Macro area 4: Measuring mitigation

One of the aspects that the interviews aim to clarify (also for the continuation of short-term activities for the 2021 sports season) is the effectiveness of the mitigation measures provided by the national and international protocols, which create the so-called "bubble" system for staff, athletes, and organizers. The experts' testimonies have been more than positive in this regard:

"I believe that the current measures are sufficient to protect health and contain the spread of infections". (I-11)

"The measures contained in the protocols have allowed us to carry out the event in safety, despite having to incur higher costs". (I-15)

The measures adopted during the 2020 season, made it possible, in principle, to stage the events safely. These results are, however, closely related to the type of race performed and the general pandemic situation. While, in the period from August to September, the races (even in stages) took place regularly without encountering any particular problems, the month of October saw the progressive reappearance of infections, which in turn created several problems. At the Giro d'Italia, several riders had to abandon the race and the bubble system came under considerable strain. Other historical races such as the Paris-Roubaix did not take place at all given the serious level of infections in the affected countries.

There was also criticism of some aspects relating to the relaunch of events and the way this was managed: various subjects (organizers, team staff, and institutional subjects) complained about a lack of coordination with regard to certain protocols and the fragmentation of the provisions set out by the local authorities:

"The measures adopted nationally and internationally were found to be suitable, however, absolute cooperation and coordination between the authorities of the various regions, or nations, is necessary ( . . . )" (I-9)

Macro area 5: Future applications

The last macro area turns our gaze toward a long-term scenario. The historic moment we are currently experiencing represents an epochal transition which, as such, is the bearer of potentially irreversible changes. The most challenging research question for this entire

work is to trace a possible evolutionary path in the world of cycling and SEs after COVID-19. The interviews show a strong awareness among stakeholders of the possible need to continue with some measures even after the pandemic has passed:

> "The risk of blocking activities for the team has led to maximum observance of the rules, I believe that some measures and attentions adopted will remain in the future". (I-13)

The measures introduced to contain the spread of the virus could become a tool or, more generally, a different systemic approach to the organization of events with greater spaces, providing more privacy for athletes during the competition:

> "I think it is appropriate that some measures remain in the future as it is right to leave the athletes some privacy before competitions". (I-15)

An opinion shared by most interviewees is that there will be an irreversible shift toward the computerization and digitization of most organizational procedures (registration management, accreditation, delivery of race packages). These tools and processes, which were already in place for several events, were implemented during the pandemic and will most likely become the future standard.

More details about the categorization are shown in Table 3.

**Table 3.** Categorization and anchor examples.

| Macro Area | Anchors | Grade of Sharing |
|---|---|---|
| Type of event | "Stage races require greater measures and controls, compared to the past the athletes were isolated with single rooms and all the equipment was continuously sanitized". (I-13); | I: 11, 9, 15, 12 |
| | "The division into areas of the site of race, however, excessive responsibilities cannot be placed on the organizer outside these spaces: start area, finish area". (I-2) | I: 9, 15, 10, 7, |
| Level of practiced activity | "I unfortunately believe that youth cycling, which does not have the same resources as professionalism, will be heavily penalized by the reduction in funds and the number of racing". (I-10) | I: 2, 9 |
| | "Cycling is a popular sport that brings champions close to the people, this element must not be lost even in this harsh reality that Covid-19 requires us and, above all, in the future". (I-15) | I:11, 3, 15, 13 |
| | "I am in favor of alternative event formulas, at an amateur level, but without distorting the typical and most exciting elements of cycling as a sport". (I-7) | I: 1, 2, 3, 7, 14, 5 not shared: I: 8,10 |
| | "I believe that sports tourism is a consequence natural of the sporting event, we must not underestimate but rather seek strongly, the link and synergies with local administrations". (I-14) | I: 7, 10, 8, 1, 4, 9, 14 |
| | "I believe that the development of sports tourism is closely linked to the adequacy of accommodation facilities and tourist staff (... It makes little sense to create routes without qualified and dedicated technical staff.)" (I-9) | I: 9, 15, 10 |
| Stakeholders | "We experienced dramatic moments during the lockdown, cycling teams get the most of their income from sponsorships and in a time of crisis these are the first costs that companies cut, especially if they are not able to generate significant visibility". (I-12) | I: 3, 5, 8, 9, 13, 12 |
| | "The absence, or limited spaces, of the exhibition and meeting areas within the events, due to the rules of distancing, I believe it will lead?? to a cut in sponsorships given the lower visibility offered, which must always be guaranteed as much as possible, and to the possibility of networking at events which cannot be completely eliminated". (I-8) | I: 4, 10 |
| | "I believe that even in the presence of limitations, teams and companies must be guaranteed opportunities for networking during events: the development of B2B relationships is fundamental for the economic resources of the sponsorship system". (I-13) | I: 4, 10, 15 |
| | "Fortunately, however, many companies remained because being linked to the sector they had a direct interest strategic to maintaining the partnership". (I-12) | I: 15, 13, 3, 8, 9, 11 |
| Measuring mitigations | "I believe that the current measures are sufficient to protect health and contain the spread of infections". (I-11); | I: 11, 2, 4, 7, 8, 9, 14, 15 |
| | "The measures contained in the protocols have allowed us to carry out the event in safety, despite having to incur higher costs". (I-15) | I: 2, 4, 5, 6, 8, 15 |
| | "The measures adopted nationally and internationally were found to be suitable, however, absolute cooperation and coordination between the authorities of the various regions, or nations, is necessary". (I-9) | I: 4, 10, 15, 3 |
| Future applications | "The risk of blocking activities for the team has led to maximum observance of the rules, I believe that some measures and attentions adopted will remain in the future". (I-13) | I: 10, 13, 15 not shared: I: 6 |
| | "I think it is appropriate that some measures remain in the future as it is right to leave the athletes some privacy before competitions". (I-15) | I: 11, 1, 13 |

## 5. Discussion

### 5.1. The Health Risk in a Cycling Race Event

The cited examples (Loncarevic et al. 2009; Blumberg et al. 2016; Rodriguez-Valero et al. 2018) show how SEs were carried out in the past despite the presence of health emergencies. This goal was reached through a complex system of risk assessment and treatment, tested and based on:

- Adequate prevention and information to stakeholders;
- Effective coordination of resources;
- A responsive scheme of alerting and tracking suspected and positive cases with a view to isolating infected people in specific and equipped areas at the event sites.

However, it is not easy to apply these solutions to a cycling race, an outdoor event which is dynamic and itinerant in nature and is largely characterized by exposure to multiple risk factors that may affect its regular performance, from meteorological (Dawkins and Stern 2004) and social factors (protests, blocks, barriers) to political ones (authorizations denied by relevant authorities) (Moran 2001).

The interviews showed that, despite the organizers' efforts, contagion can never be a zero-risk. The bubble system, however, which was conceived as an applicator by UCI, allowed the races to be carried out in acceptably safe conditions. This can reasonably lead us to think that this scheme may well represent the starting point for the development of anti-pandemic protocols in cycling events. The organizing team of a cycling event must provide an effective response plan, which can be activated in case of an emergency so that anti-pandemic countermeasures are implemented and the effects of this unexpected event are minimized. While it is certainly difficult, and sometimes impossible, to cope with situations that we define as unpredictable, models and mechanisms may be developed to deal with situations that have little chance of manifesting themselves but still represent an eventuality.

The current organizational management model always provides alternative solutions to ensure the smooth running of competitions, although reality can place the organizers in a difficult situation at any time.

### 5.2. The Effects of SEs on Tourism

Today, international tourism is among the economic sectors most seriously affected by COVID-19 (Kyrylov et al. 2020). Sport and tourism are a very important economic driver for destinations and their resident communities (Borovcanin et al. 2020). The creation and organization of a high-level sporting event cannot be separated from the deep involvement of a destination and its institutional structure (Masterman 2009). The locational attachment of an event is an element that must guide all managerial, organizational, and above all, promotional aspects. The creation of an image/brand linked to a sporting competition must transmit values and a message that are in keeping with the characteristics of the social context of reference. The more consistent the process, the greater the potential success of the event.

When it comes to sport tourism, cycling is a good discipline that is well suited to promoting destination tourism. Its peculiarity as an outdoor sport that moves and lives within the destination contains enormous potential, which, if well exploited, can lead to exponential growth in the reference community.

These considerations are valid both for professional cycling and for the amateur movement. The major world events of the UCI World Tour circuit have a fascination and appeal mainly linked to the technical aspect and spectacular atmosphere. The real attraction for a fan/tourist is the opportunity to see the athletes in action along the course and at the same time to have the chance to visit a new destination. In this context, it is possible to create more direct benefits for the territory than just destination branding. The major events that bring together thousands of people from all over the world in the Alps are an excellent example of how sport can promote and grow the area, all in full respect of the environment.

The COVID-19 pandemic has changed the patterns and practices adopted by the tourism industry, policymakers, and practitioners, who are being forced to develop new crisis-readiness mechanisms to counteract the current pandemic as well as possible future pandemics. Private and public policy support must, therefore, be coordinated to guarantee capacity building and operational sustainability of the travel tourism sector during the initial post-COVID-19 period (Škare et al. 2021).

### 5.3. The Dynamic Perspective of Results

The content, evaluations, and opinions emerging from the interviews constitute a solid starting point for answering the research questions that constitute the main coordinates of the work at hand. The field survey, however, which gave a voice to the experts by analyzing the main anti-contagion measures applied and the related criticalities, merely represents a static photograph. To better contextualize the information, it is useful to evaluate the possible interrelationships in a "dynamic" guise, so that it may be implemented in a certain way.

To this end, the five thematic areas, used as a "matrix" for coding the content of the interviews, were related to the main anti-COVID-19 measures as application variables. The aim of this process is to investigate the degree of effectiveness of the individual mitigation measures proposed.

Table 4 illustrates the logical-thematic relationships:

**Table 4.** Dynamic relationships between anti-covid measures and analysis factors.

| Variables | | Anti-Covid Measures Adopted or Proposed | | | | | | |
|---|---|---|---|---|---|---|---|---|
| | | Reduced and Controlled Access to Areas | New Health Checks for Athletes and Staff | Minimum Contact with Supporters | Digitization of Organizational Processes | Logistics of Transfers and Accommodation Services Monitored | Limited Number of Starters | New Event Formats |
| Type of event | Grand tours | x | x | x | x | x | | |
| | One-day races | x | just once, not repeated | x | x | | | x |
| | Multi-day events on the same site | x | x | x | x | x | | x |
| Level of activity | Amateur | x | | | x | can change, but the amateur competitions are mostly suspended | x | x |
| | Professional | x | x | x | x | x | | |
| | Youth categories | x | x | x (necessary measure, but very limited need) | x | x | | |
| | Tourist | not always necessary | | | x | | | x |
| Future application | Yes | | x | x | x | | | x |
| | No | | | | | | | |
| Stakeholders acceptance | Elevated | x | x | | x | x | | x |
| | Significative | | | x | | | | |
| | Minimum | | | | | | x | |
| Impact on organizational costs | Elevated | x | x | | | x | | |
| | Significative | | | x | | | | x |
| | Minimum | | | | x | | x | |
| Impact on containment of contagion | Elevated | x | x | x | x | x | | x |
| | Significative | | | | | | | |
| | Minimum | | | | | | x | |
| Introduction | Already present | x | x | x | x | x | | |
| | Not yet introduced | | | | | | | |
| Effect on sponsors and business partners | Elevated | x | | x | | | x | |
| | Significative | | | | | | | |
| | Minimum | | x | | x | x | | x |

From the comparison, we highlight several considerations regarding possible anti-contagion measures below:

### 5.3.1. Reduced and Controlled Access to Areas

The regulation of access and the division of the event site into separate areas were among the first, and most decisive, measures adopted to avoid the spread of the virus in the sports event scene. Experts accepted these measures, despite the consequent increase in organizational costs. The measures were found to have had an excellent impact on the containment of infections. It is difficult to conceive that the stringent access regulations will become permanent once the pandemic has ended. It is very likely, however, that in the event of future emergencies they will become established as a practice. The measures had to be applied transversally with all events having to comply.

### 5.3.2. New and Repeated Checks on Athletes and Staff

The rigid bubble system, prepared by the UCI as a standard protocol, made it possible to monitor the health of the athletes and team staff, ensuring—with the exception of some foreseeable positive cases—the regular running of the competitions. Its application is currently limited to professional athletes only. The need for repeated checks is greater in the case of stage races, where in addition to the creation of the bubble, its impenetrability must also be guaranteed.

### 5.3.3. Limited or Inhibited Contact with the Public

The drastic elimination of the spaces in which fans and athletes interact (Bond et al. 2020; Mastromartino et al. 2020) was both necessary and indispensable; the measure was shared and applied without encountering any problems. This need is less pronounced in youth or second tier competitions where the presence of the public is felt to a lesser degree. By its very nature, cycling is a sport that brings people and fans closer together. The measures, though generally accepted—and also confirmed by the interviewees—would not have any meaning in a post-COVID-19 context. In fact, general concern was expressed by the experts that this factor could distance fans and enthusiasts from cycling to the detriment of visibility and interest in this discipline, leading to a loss of important financial resources for the whole system (sponsorships, commercial partnerships etc. . . . ).

### 5.3.4. Digitization of Organizational Processes

The use of new technologies for the management of different event phases has proved to be of transversal benefit and one that was supported by all stakeholders. The simplification or elimination of certain activities to be carried out at the event site (secretariat, registrations etc. . . . ) made it possible to create safe gatherings.

### 5.3.5. Transfer and Accommodation Logistics Monitored

Especially on multi-day journeys, transfers and logistics activities have become much more complex and expensive. It is unlikely that the measures adopted in this regard will remain in the future, unless they are strictly necessary.

### 5.3.6. Limitations on the Number of Starters

On a professional level, this measure was neither envisaged nor deemed necessary. It will probably gain more traction in the context of amateur competitions (in the first phase of recovery) where athletes turn out in much more significant numbers.

### 5.3.7. New Event Formats

A widespread desire among the interviewees is to take this pandemic as an opportunity to relaunch amateur cycling by adopting an approach more oriented toward sports tourism. Where, on the one hand, the inability to carry out competitions has led to a natural contraction in the number of events and subscribers, various organizers, have

reinvented their events by introducing different competitive formats (individual time trials, for example) or simple cycles. Meetings are now taking place in compliance with social distancing rules. The pandemic has also become an exceptional catalyst for the rediscovery of the "half" bicycle which today is as functional, ecological and safe as ever.

### 5.4. Managerial Implications

Based on our research output, we can highlight several managerial implications that should be considered by every sport event professional:

- New expert positions and responsibilities among the organizational staff. These experts are charged with managing an unprecedented type of risk that requires an adjustment and rethinking of the processes of conception, planning, and management of SEs.
- A new concept and structuring of the competition site and the regulation of access might be adopted. Re-thinking logistics and space management, from a more rational and controlled perspective, could continue even after the pandemic. One limitation may be the lack of global standardization of the rules and their timely application. This could allow SEs to be carried out safely, while reducing the extent of infections.
- A "smart" strategy for stakeholder engagement must be chosen and improved: interest in cycling on the part of commercial and technical partners has not faded despite the severe difficulties that companies have had to face due to the virus. However, a new sponsorship and television rights system could be developed, since visibility remains the main attraction for a company sponsoring cycling events. If this fails due to the cancellation of competitions, or a reduction in the audience reached through the media, there is a risk of losing competitiveness in relation to other sports and, therefore, financial resources and investments.
- Even at the amateur and cycling-tourism level, the difficulty of managing traditional competitions during the intermediate phase of COVID-19 forced organizers to explore new solutions. The feedback received from practitioners was very good. The future perspective for the amateur cycling movement should be more inclusive in scope with a re-evaluation of the recreational aspects. Competition should be neither eliminated nor demonized but evaluating "non-competitive" formulae can create value and satisfaction in relation to the impact of a cycling event on amateur athletes. These two souls could coexist within the same event. In terms of promoting a destination among tourists, this aspect could also be considered a winning factor. Following this scheme in the event of future pandemics, it will be easier to switch to a different event formula and save the SE. The future goal for cycling event managers is to create a new and resilient model which, by adopting these measures based on the previous experience of COVID-19, will be able to respond to future health emergencies with a limited impact on stakeholders, athletes and staff.

Future research will need to consider the effects of this pandemic in redefining the planning and management of risks at SEs.

To this end, it will be useful to update the meaning of risk, and also to understand the real economic, social, and environmental impact on SEs, on the context in which they take place, and on tourism destinations.

### 6. Limitations

The present study provides an interesting perspective of the role of SEs in cycling in the post-pandemic era. However, we would like to highlight several limitations. The sample interviewed comprised participants mainly from Italy. In order to increase the reliability of the results or to make comparisons with other organizational models, the study should be replicated with a different sample. Second, the results should be replicated in different sports disciplines to provide research and management with a wider range of results and implications. Finally, we must consider that the Covid-19 pandemic has had a strong impact on tourism. Current restrictions around the world have prevented travel which may consequently influence attitudes to travel and participation in SEs. Attitudes toward

cycling SEs might be influenced by new factors and behaviors and thus become new stimuli for future research.

### 7. Conclusions and Future Research

This work had the primary purpose of providing an analytical interpretation of a new phenomenon with unknown effects in the world of sports with a view to identifying the possible scenarios and characteristics of cycling SEs in the post COVID-19 era.

The analysis of the literature showed how, in the past, major SEs have successfully faced health emergencies which, however, have never matched the global challenges posed by COVID-19.

Thus, by examining the measures adopted to counter the COVID-19 contagion, staff involved in the organization of SEs must essentially take note of the emergence of new approaches and responsibilities. These subjects faced an unprecedented type of risk that required a fundamental adjustment and a rethink of the processes of conception, planning, and management of SEs. A new concept and structure for the competition site and the regulation of access could be adopted: adjusted logistics and space management that incorporates a more rational and controlled perspective, could continue even after the pandemic. The creation of a new and specific international standard for the management of competition sites could be a very effective solution in the event of a new pandemic. In any part of the world, standardized rules and their timely application could allow events to be carried out safely, while reducing the extent of infections. The "bubble system" protocol developed by UCI for the relaunch of professional races has proved to be suitable and effective for the safe conduct of the events and in the case of future emergencies, it will probably be refined and re-applied alongside the management rules on the competition site.

The interest in cycling on the part of commercial and technical partners has not faded despite the severe difficulties that companies have had to face due to the virus. However, a rethink of the sponsorship and television rights system would be well advised, since visibility remains the main attraction for a company sponsoring cycling. If this fails due to the cancellation of competitions, or due to a reduction in the potential audience reached through the media, there is a risk of losing competitiveness with respect to other sports and therefore financial resources and investments.

The spaces in which fans and athletes can interact will probably be reduced, though not completely eliminated. The fear of health risks to athletes, with the consequent economic and sporting damage to the entire team, has reached a different level of sensitivity among organizers and team managers. However, in the future, when the current pandemic emergency is finally over, various prescriptive measures will no longer be necessary.

In the amateur sector and cycling-tourism, new concepts should be adopted. The impossibility of being able to compete as before has forced the organizers to explore different organizational methods and solutions, including competitive and non-competitive ones. The future outlook facing amateur cycling cannot be separated from the re-evaluation and strengthening of these aspects. Competition must not be eliminated or demonized, but a non-competitive formula may also create value and satisfaction around an amateur cycling event. These two souls might well be able to coexist within the same event.

Regarding the promotion of a tourist destination, this option could be considered a winning card. The agonist "feels" the event by focusing on his/her sporting performance, while being more likely to build loyalty to the event and the destination. An amateur who exclusively perceives the event as pure enjoyment and fun, will more likely look at the destination superficially.

**Author Contributions:** Conceptualization and writing—original draft preparation, F.B. and N.M.; methodology and supervision, N.M. and E.H.; editing, N.M. and F.B.; data, N.M. and P.A.S.; writing—review, P.A.S., E.H. and F.B. All authors have read and agreed to the published version of the manuscript.

**Funding:** This research received funding from Vice Rector of Research of the University of Innsbruck—Austria.

**Institutional Review Board Statement:** Not applicable.

**Informed Consent Statement:** Not applicable.

**Data Availability Statement:** The data presented in this study are available on request from the corresponding author.

**Acknowledgments:** We would like to thank the Editor and the Referees for their constructive support that make this paper more valuable, and the Vice Rector of Research of the University of Innsbruck—Austria.

**Conflicts of Interest:** The author declares no conflict of interest.

## Appendix A

**Table A1.** Overview of the Interview Guideline.

| | **Section 1.** Event management |
|---|---|
| 1 | In your opinion, how will logistics and accreditation choices change for major events in the post-Covid era? |
| 2 | What measures do you think are appropriate to ensure the health of athletes and all practitioners? |
| 3 | Will the situation generated by the pandemic lead, in your opinion, to a limitation of the number of participants, staff and companions admitted to the events, as a standard protocol for cycling? |
| 4 | How do you think the prospect of a possible pandemic event capable of blocking the whole system will affect the future assessments of stakeholders (sponsors and other commercial partners)? |
| | **Section 2.** Risk management |
| 5 | How do you think the approach to risk management in sporting events will change after covid-19? |
| 6 | Which tools used in the past could be useful and which ones could have new applications? |
| 7 | Do you think that, given the possibility of a new pandemic in the future, is it necessary to include health risk management as a priority for future sporting events? |
| | **Section 3.** Sport tourism |
| 8 | How is the impact of a sporting event such as a cycling race on the territory planned and managed from a tourism point of view ? |
| 9 | How important do you think the legacy of a sporting event is for an organizer gauging the success of the event? |
| 10 | What measures do you think are strategic to relaunching cycling tourism (sports and recreational) after covid-19? |

## Appendix B

**Table A2.** Chart of Stakeholders Interviewed.

| ID Number | Professional Area | Qualification Assigned/Role | Practice Level | Date | Duration | Contacted By |
|---|---|---|---|---|---|---|
| I-1 | Organization management | Manager with qualified experience in the sector | Amateur | 16 June 2020 | 1 h 8 min | Microsoft teams call |
| I-2 | Institutional | Unit director | Youth categories and amateur | 25 June 2020 | 47 min | Skype call |
| I-3 | Technical | Coach | Youth categories and amateur | 8 July 2020 | - | Written compilation |
| I-4 | Institutional | Top manager | Amateur | 1 July 2020 | 33 min | Skype call |
| I-5 | Tourism | Chief Executive Officer of a company operating in the sector | Amateur | 14 June 2020 | 51 min | Skype call |
| I-6 | Academic | Professor with qualified professional experience in the areas covered by the study | Youth categories and amateur | 22 June 2020 | 20 min | Skype call |
| I-7 | Organization management | Organizing committee member | Amateur | 19 June 2020 | 51 min | Zoom call |
| I-8 | Organization management | President of the organizing committee | Amateur | 19 June 2020 | 53 min | Zoom call |

**Table A2.** *Cont.*

| ID Number | Professional Area | Qualification Assigned/Role | Practice Level | Date | Duration | Contacted By |
|---|---|---|---|---|---|---|
| I-9 | Organization management | Director and manager responsible for the event | Youth categories and amateur | 18 July 2020 | 42 min | Skype call |
| I-10 | Organization management | Sports club and organizing committee President | Amateur | 18 July 2020 | 1 h 8 min | Skype call |
| I-11 | Technical | Ex-pro rider | Professional | 13 August 2020 | 15 min | WhatsApp video call |
| I-12 | Institutional | Area manager | Professional | 21 August 2020 | - | Written compilation |
| I-13 | Marketing and communication | Area manager | Professional | 16 September 2020 | 28 min | Skype call |
| I-14 | Organization management | President of the organizing committee | Amateur and Professional | 21 September 2020 | 49 min | Skype call |
| I-15 | Organization management | Director and organizational manager | Professional | 2 October 2020 | 26 min | Phone call |

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
