# Peer review of "How the Covid-19 Pandemic Influenced the Approach to Risk Management in Cycling Events"

_jrfm, doi:10.3390/jrfm14070296_

Round 1

Reviewer 1 Report

A good job. When studying a phenomenon with little prior background, an inductive and exploratory method has been chosen, using qualitative techniques. Very good choice.

Qualitative research is not very reproducible, therefore, it is necessary to give a much greater effort to be able to describe the methodology and the protocol used. In this sense, it is necessary that the authors expand on the methods section.

They should talk about the research validation process: validation of the interview and the researchers (content and construct). These investigations require prior training and experience and it is always advisable to do a pilot interview to improve the script.

It is necessary to indicate if the interviews were carried out by the same researcher or if they changed (a fact that may affect the validity that I comment in the previous paragraph).

Likewise, it is advisable to use a software for the tabulation and categorization of the responses. How was this process developed?

With these indications the study will acquire more robustness and reproducibility, improving its quality.

Author Response

Reviewer 1

1

A good job. When studying a phenomenon with little prior background, an inductive and exploratory method has been chosen, using qualitative techniques. Very good choice.

Thank you for this general comment and for all the comments to which we have tried to respond, including the revision of the manuscript.

2

Qualitative research is not very reproducible, therefore, it is necessary to give a much greater effort to be able to describe the methodology and the protocol used. In this sense, it is necessary that the authors expand on the methods section.

In the method section (p. 10-11), we have tried to describe with greater accuracy the protocol used to decode the interview outputs to achieve the findings of the analysis. We have rearranged the relevant part of section 3.2 and have added new content.

3

They should talk about the research validation process: validation of the interview and the researchers (content and construct). These investigations require prior training and experience and it is always advisable to do a pilot interview to improve the script.

We agree and thank you for the important suggestion which served to improve the whole of section 3.2. Research setting, participants, and procedure.

This should make this part easier to understand, especially the process of validating the research. The absence of a pilot interview has been included as a limitation. We agree that this step would have further improved the validation of the interviews.

4

It is necessary to indicate if the interviews were carried out by the same researcher or if they changed (a fact that may affect the validity that I comment in the previous paragraph).

Thank you for this observation. In order to avoid bias or otherwise affect validation, only one researcher managed the interviews.

5

Likewise, it is advisable to use a software for the tabulation and categorization of the responses. How was this process developed?

We agree with this suggestion. In this specific case, Microsoft Excel was used to categorize the interview responses. However, the use of specialized software would have further improved the categorization process.

6

With these indications the study will acquire more robustness and reproducibility, improving its quality.

We agree.

Reviewer 2 Report

The article is interesting because it will help managers and organisers of cycling events to know how to apply risk management in the organisation of these events, focusing on the measures that should be introduced to deal with health emergency situations in cycling events, through the comparison of the measures that have been taken in similar events, to help them to evolve their own events.

In general, the article is well developed, but for a better understanding of the results of the qualitative study, it would be interesting to indicate in the methodology part the profile of the people interviewed (although they expose it a little in the annex through a table): level of the company they belong to (national or international), country they belong to, member of the organising committee of a federation, company, club, etc.

Author Response

Reviewer 2

1

The article is interesting because it will help managers and organisers of cycling events to know how to apply risk management in the organisation of these events, focusing on the measures that should be introduced to deal with health emergency situations in cycling events, through the comparison of the measures that have been taken in similar events, to help them to evolve their own events.

Thank you for this positive evaluation and the feedback on how to further improve our manuscript.

2

In general, the article is well developed, but for a better understanding of the results of the qualitative study, it would be interesting to indicate in the methodology part the profile of the people interviewed (although they expose it a little in the annex through a table): level of the company they belong to (national or international), country they belong to, member of the organising committee of a federation, company, club, etc.

Thanks for bringing this up. We have re-phrased the sentence in the Methodology section (p.10), adding a more detailed description of interview profiles.

Reviewer 3 Report

This study explores the risk management in cycling events through three research questions, related to measures introduced in cycling events, differences in measures with previous events, and the possible evolution in the post-covid era. The topic of the paper is potentially interesting and attractive for professionals in the sport field, especially event organizers. The paper fits within the scope of the journal. However, I consider that the manuscript should be improved in some ways:

The aspect that concerns me the most is related to the description of the Methods:

  • Where was the research performed? Italy? Since the manuscript is based on a qualitative design, this piece of information is essential.
  • How do the authors justify the interview I-3 (written compilation)? I have strong concerns about how this interview could be considered in the same way as the others. How was the written interview analyzed? How could this interview interfere with the results?
  • In the same line, how the different modes of contact: with sound and image (e.g. Zoom, Skype, etc.), or only sound (e.g. phone call) can affect the results?
  • The authors state that fifteen experts in the field of cycling events were interviewed. As this is a qualitative study, the sample size could be acceptable. However, did the authors reach the saturation point? (Point where new informants do not reveal new information).

Specific comments

In the Introduction section (page 2), the authors state that “Although more research has been carried out into the security governance of SEs in recent years (Giulianotti and Klauser 2010; Boyle and Haggerty 2012)”. Could you please provide more recent research (preferably in the last 5 years, 2017-2021)?

In the Theoretical Background section, at the Literature Review, some ideas about the legacy of sport events are presented (page 3). I am not sure how well this information fits in this article. I kindly suggest the authors better introduce and link these paragraphs with previous or following ideas.

At the beginning of the Findings, the sentence “There was no lack of discrepancies in the opinions of the interviewees, as well as shared measures and actions” is understandable to somewhat point, but could be clearer. Please rewrite the sentence in a clearer way.

Also in the first paragraph of Findings, I believe the information provided in “Starting from such a context, this work aims to investigate and explore possible future scenarios. We adopted a deductive categorization system and, on the basis of the answers, a number of responses was analyzed for each category (Mayring 2010)” is more suitable in the Methods section than in the Findings. Please consider the possibility to rearrange this information within the article.

The comments are expected to be useful.

Author Response

Reviewer 3

1

This study explores the risk management in cycling events through three research questions, related to measures introduced in cycling events, differences in measures with previous events, and the possible evolution in the post-covid era. The topic of the paper is potentially interesting and attractive for professionals in the sport field, especially event organizers. The paper fits within the scope of the journal.

However, I consider that the manuscript should be improved in some ways:

The aspect that concerns me the most is related to the description of the Methods:

·       Where was the research performed? Italy? Since the manuscript is based on a qualitative design, this piece of information is essential.

·       How do the authors justify the interview I-3 (written compilation)? I have strong concerns about how this interview could be considered in the same way as the others. How was the written interview analyzed? How could this interview interfere with the results?

·       In the same line, how the different modes of contact: with sound and image (e.g. Zoom, Skype, etc.), or only sound (e.g. phone call) can affect the results.

·       The authors state that fifteen experts in the field of cycling events were interviewed. As this is a qualitative study, the sample size could be acceptable. However, did the authors reach the saturation point? (Point where new informants

do not reveal new information).

We appreciate this general comment. We have tried to take on board all comments and suggestions to further improve the manuscript.

Thank you for this very useful remark; it is fundamental to identify the geographical research area. On page 10, we have included a new and specific sentence on this point.

Interview I-3 was acquired in written form because, for the interviewed agenda, it was very difficult to find a sufficient time slot. We considered and analyzed this interview in the same way as the others because the participation and the answers given provided precise and rich data for the study.

We also spoke to the interviewee on the phone to verify that the answers had been interpreted correctly. We do not think that this method of interviewing could represent a bias that would invalidate the overall results of the research.

In general, the feedback from the interviews was of a very good quality, but we have not noticed any major differences and, above all, it appeared to us that the final result was actually satisfactory in terms of moderation of the interviews and content. To explain this better, we have included a new sentence in the Methodology section.

We think there is no bias given the different tools used in collecting the interviews (Creswell and Creswell, 2018). However, limitations remain in the fact that in one-on-one interviews the presence of the interviewer may represent a bias and that not all interviewees are guaranteed to be balanced and responsive. We have added this in the limitations section.

As stated in the pre-review version of the manuscript, we confirm that the saturation point was reached. The research sample had a good grade of reliability. However, we have been more specific in section '3.2. Research setting, participants, and procedure' also with reference to saturation, as stated in Glaser and Strauss (1967) and Guest, Bunce, and Johnson (2006).

2

Specific comments

In the Introduction section (page 2), the authors state that “Although more research has been carried out into the security governance of SEs in recent years (Giulianotti and Klauser 2010; Boyle and Haggerty 2012)”. Could you please provide more recent research (preferably in the last 5 years, 2017-2021)?

In the Theoretical Background section, at the Literature Review, some ideas about the legacy of sport events are presented (page 3). I am not sure how well this information fits in this article. I kindly suggest the authors better introduce and link these paragraphs with previous or following ideas.

At the beginning of the Findings, the sentence “There was no lack of discrepancies in the opinions of the interviewees, as well as shared measures and actions” is understandable to somewhat point, but could be clearer. Please rewrite the sentence in a clearer way.

Also in the first paragraph of Findings, I believe the information provided in “Starting from such a context, this work aims to investigate and explore possible future scenarios. We adopted a deductive categorization system and, on the basis of the answers, a number of responses was analyzed for each category (Mayring 2010)” is more suitable in the Methods section than in the Findings. Please consider the possibility to rearrange this information within the article.

Thanks for the suggestion. We have added more scholars: Ludvigsen & Parnell, 2021; Whelan & Molnar, 2018.

We absolutely agree that the legacy part of this section should be better linked to the focus theme of the manuscript. We have revised this part on p. 4.

In the reviewed text, we have rewritten the sentence so that it is clearer and easier to understand.

Thank you for the appropriate remark; in the new version of the document we have located the information in the overview paragraph of the methodology section, from pages 9 to page 11.
